# Quantification of the impact of hydrology on agricultural production as a result of too dry, too wet or too saline conditions

5   M.J.D. Hack-ten Broeke[1], J.G. Kroes[1], R.P. Bartholomeus[2], J.C. van Dam[3], A.J.W. de Wit[1], I. Supit[1]
D.J.J. Walvoort[1], P.J.T. van Bakel[4], R. Ruijtenberg[5]

[1]Alterra, Wageningen University and Research Centre, PO Box 47, 6700 AA Wageningen, the Netherlands
10  [2]KWR Watercycle Research Institute, PO Box 1072, 3430 BB Nieuwegein, the Netherlands
[3]Wageningen University, Soil Physics and Land Management group, PO Box 47, 6700 AA Wageningen, the Netherlands
[4]De Bakelse Stroom, Simon Vestdijkstraat 15, 6708 NW Wageningen, the Netherlands
[5]STOWA, postbus 2180, 3800 CD Amersfoort, the Netherlands

15  *Correspondence to*: M.J.D. Hack-ten Broeke (mirjam.hack@wur.nl)

**Abstract**

For calculating the effects of hydrological measures on agricultural production in the Netherlands a new comprehensive and climate proof method is being developed: WaterVision Agriculture (in Dutch: Waterwijzer Landbouw). End users have asked for a method that considers current and future climate, that can quantify the differences between years and also the effects of extreme weather events. Furthermore they would like a method that considers current farm management and that can distinguish three different causes of crop yield reduction: drought, saline conditions or too wet conditions causing oxygen shortage in the root zone.

WaterVision Agriculture is based on the hydrological simulation model SWAP and the crop growth model WOFOST. SWAP simulates water transport in the unsaturated zone using meteorological data, boundary conditions (like groundwater level or drainage) and soil parameters. WOFOST simulates crop growth as a function of meteorological conditions and crop parameters. Using the combination of these process-based models we have derived a meta-model, i.e. a set of easily applicable simplified relations for assessing crop growth as a function of soil type and groundwater level. These relations are based on multiple model runs for at least 72 soil units and the possible groundwater regimes in the Netherlands. So far, we parameterized the model for the crops silage maize and grassland. For the assessment, the soil characteristics (soil water retention and hydraulic conductivity) are very important input parameters for all soil layers of these 72 soil units. These 72 soil units cover all soils in the Netherlands. This paper describes i) the setup and examples of application of the process-based model SWAP-WOFOST, ii) the development of the simplified relations based on this model and iii) how WaterVision Agriculture can be used by farmers, regional government, water boards and others to assess crop yield reduction as a function of groundwater characteristics or as a function of the salt concentration in the root zone for the various soil types.

**1 Introduction**

The United Nations formulated 17 Sustainable Development Goals (SDG) for the period 2015-2030 (http://sustainabledevelopment.un.org/focussdgs.html). Prominent goals are 'End hunger, achieve food security and improved nutrition and promote sustainable agriculture' (SDG 2) and 'Ensure availability and sustainable management for water and sanitation for all' (SDG 6). A key factor to achieve these goals is efficient use of water in agriculture. Currently agriculture uses 92% of the global fresh water use, exceeding by far the use by industry or households (Keesstra et al., 2016; Hoekstra and Mekonnen, 2012). We may release large amounts of water for extra food production or other pressing human or natural needs by increasing the water productivity in agriculture. However, this requires a profound knowledge of the effects of dry, wet, and saline conditions on growth and yield of agricultural crops.

The changing climate and weather conditions aggravate the need for reliable tools to assess crop yields in view of water stresses. Furthermore increased water extraction from aquifers, deteriorating water quality and rationed water supply and irrigation services are some reasons for increased agricultural drought stress in arid and semi-arid regions. Over the past 60

years, soil water conditions have been generally wetting over the western hemisphere and drying over the eastern hemisphere, mostly in Africa, East Asia and Europe. Trends over the past 20 years indicate intensification of drying in northern China and southeast Australia, and switches from wetting to drying across much of North America, and southern South America, in part because of several large-scale and lengthy drought events (FAO, 2015). Hotspots of pressures on soil

water quantity and quality are e.g. the North China Plain, Australia, the southwestern United States and Middle East and North Africa (MENA) region. The World Bank took the initiative to generate an improved understanding of water issues in the MENA region, including associated marginal cost of water supply to meet the growing water need. Unmet demand for the entire MENA region, expressed as percentage of total demand, will increase from 16% currently to 37% in 2020-2030 and 51% in 2040-2050. A large number of measures were evaluated to meet the demand. The measure with the highest

benefit/cost ratio is increase of agricultural water productivity, which requires proper attention for both drought and waterlogging risks during crop production (FutureWater, 2011).

Although salt-affected soils are widespread and an increasingly severe problem, no accurate recent statistics are available on their global extent (FAO, 2015). The best available estimates suggest that about 412 million ha are affected by salinity and

618 million ha by sodicity (UNEP, 1992), but this figure does not distinguish areas where salinity and sodicity occur together. The Soil Map of the World (FAO/UNESCO, 1980) depicted a similar extent of 953 Mha affected by salinity (352 million ha) and sodicity (580 million ha). According to Jones et al. (2012) excess levels of salts in soils are believed to affect around 3.8 million ha in Europe. Naturally saline soils can be found in Spain, Hungary, Greece and Bulgaria. As an effect of irrigation artificially induced saline soils occur in Italy, Spain, Hungary, Greece, Portugal, France, Slovakia and Romania.

In the Netherlands salinization is not caused by accumulation of salts, but occurs in dry summers when crops are irrigated with salt-rich water from the ditches in the western part of the country near the sea. Excess rainfall in the winter period will always wash away these excess salts from the root zone in these areas. Yet, these conditions negatively affect crop production.

Models on soil hydrology and crop growth evolve and both integrate and simulate not only the natural interactions but also the effect of farm management decisions. As a consequence the currently used instruments for quantifying the effect of hydrological conditions in the root zone on agricultural production for instance in the Netherlands are no longer sufficient. Different groups of users, like water boards, provinces, drinking water companies and the National Department of Waterways and Public Works are therefore demanding an instrument that can determine crop yield effects as a result of

drought, too wet or too saline conditions for both current and future climatic conditions. In order to be applicable to future climate conditions the system has to be based on process based models; implicit incorporated expert knowledge cannot be extrapolated to unknown conditions. Other important specifications are that the results must be reproducible, that farm management is included, and that new insights can be added in the future.

WaterVision Agriculture should become that new instrument, based on linked model simulations for soil hydrology (SWAP) and crop growth (WOFOST) on the basis of different weather conditions and future climate. Plant growth is determined by the availability of solar radiation, $CO_2$, water, oxygen and soil nutrients. To achieve maximal growth plants always try to take up sufficient water and oxygen from the soil. When the availability of water (too dry) or oxygen (too wet) in the root

zone is insufficient, plants experience either drought or oxygen stress. When the salt concentration in soil water is too high, the water uptake will also decrease.

In WaterVision Agriculture the agrohydrological simulation model SWAP (van Dam et al., 2008) and the crop growth simulation model WOFOST (van Diepen et al., 1989) together form the core of the calculation of crop yields as a function of

soil moisture conditions. We have linked these models on a daily basis to ensure realistic interaction between water in the root zone and crop growth. For instance: dynamic root growth as a function of weather and soil conditions instead of assuming a static rooting depth will influence water uptake and yield reduction, caused by drought or oxygen stress. This will reduce leaf area and this in turn will reduce transpiration in a more realistic way than assuming average annual crop development. Furthermore the linkage of these models enables us to assess the effects of future climate on the interaction

between hydrology and crop growth.

Based on these complex process-based models we want to arrive at an easily applicable method with direct relationships between groundwater characteristics and crop growth. For this we have derived a meta-model, which mimics the relevant processes involved and generates roughly the same model results as the SWAP-WOFOST model would do, using much less

input data. This facilitates the practical application of scientific knowledge. In this paper we describe how this meta-model for WaterVision Agriculture was derived for grassland and silage maize and how it can be used. The ultimate project goal is to develop a comprehensive and well recognized method for quantifying agricultural effects of hydrological change.

## 2. Materials and methods

First we describe the simulation models used for WaterVision Agriculture, followed by data used for model testing. Then we describe the production of the meta-model and the data required for this action.

### 2.1 SWAP

5   The SWAP (Soil-Water-Atmosphere-Plant; Van Dam et al., 2008) model is the core of WaterVision Agriculture and is a widely used model for the determination of the actual evapotranspiration as a function of meteorological data, combined with crop and soil data (Feddes and Raats, 2004). The model simulates water flow in the unsaturated and saturated upper part of the soil profile, where the interaction between groundwater and surface water is important. The model SWAP calculates the water transport, dissolved substances and soil temperature (Fig. 1).

10  Water transport simulation is based on the Richards equation with a variable sink term for root water extraction. The potential transpiration rate depends on atmospheric conditions (air temperature, wind speed, solar radiation and air humidity) and plant characteristics (reflection coefficient, stomatal resistance, plant height and leaf area index). The potential root water extraction rate at a certain depth, $S_p(z)$ (d$^{-1}$), is considered to be proportional to the root length density and the potential transpiration rate:

$$S_p(z) = \frac{L_{root}(z)}{\int_{-D_{root}}^{0} L_{root}(z)\,dz} T_p$$

with $L_{root}$ the root length density (cm$^{-2}$) and $D_{root}$ the root layer thickness (cm).

Stresses due to dry or wet conditions and/or high salinity concentrations may reduce $S_p(z)$. The drought stress in SWAP is described by the dry part of the reduction function proposed by Feddes et al. (1978), which is depicted in Fig. 2a. In the 20  moderate pressure head range $h > h_3$ root water uptake is optimal. Below $h_3$ root water uptake linearly declines due to drought until zero at $h_4$ (wilting point). The critical pressure head $h_3$ increases for higher potential transpiration rates of $T_p$.

Oxygen stress, defined as daily respiration reduction (i.e. potential minus actual respiration) is calculated with the process-based method of Bartholomeus et al. (2008) for oxygen transport and consumption, which uses generally applied 25  physiological and physical relationships to calculate both the oxygen demand of and the oxygen supply to plant roots (Fig. 3). Oxygen stress occurs when the actual root respiration is lower than the potential root respiration, i.e. when the oxygen supply cannot meet the oxygen demand of plant roots. Root respiration is determined by interacting respiratory (i.e. oxygen consuming) and diffusive (i.e. oxygen providing) processes in and to the soil. Plant roots respire at a potential rate under optimal soil aeration and thus non-limiting oxygen availability. This potential root respiration is in equilibrium with the 30  oxygen demand of plant roots, which is determined by plant characteristics and soil temperature (Amthor, 2000) only. Upon increasingly wetter conditions, however, the gas-filled porosity of the soil decreases and oxygen availability becomes

insufficient for potential root respiration. The method of Bartholomeus et al. (2008) is applied to all soil layers of SWAP, to account for layer-specific soil physical properties, moisture contents and temperatures.

SWAP uses the response function of Maas and Hoffman (1977) for salinity stress (Fig. 2b). Below the critical concentration of $EC_{max}$ (dS/m) no salinity stress is assumed. At salinity levels above $EC_{max}$ the root water uptake declines with a constant slope of $EC_{slope}$ (m/dS). The actual root water flux, $S_a(z)$ (d$^{-1}$) is derived in SWAP by multiplication of the stress factors due to drought, oxygen and salt stress:

$$S_a(z) = \alpha_d(z)\, \alpha_o(z)\, \alpha_s(z)\, S_p(z)$$

where $\alpha_d$ (-), $\alpha_o$ (-) and $\alpha_s$ (-) are reduction factors due to drought, oxygen and salinity stress, respectively. Integration of the actual root water flux over the root zone yields the actual transpiration rate $T_a$ (cm d$^{-1}$):

$$T_a = \int_{-D_{root}}^{0} S_a(z)\, \partial z$$

Oster et al. (2012) compared five agrohydrological models (ENVIRO-GRO, HYDRUS, SALTMED, SWAP and UNSATCHEM) that simulate the effect of continually changing salinity and matric stress on crop yields. These models all assume a linear relation between relative crop transpiration and relative dry matter production. As input they used soil and climatic conditions of the San Joaquin Valley of California to simulate the yields of forage corn for various amounts of irrigation and water quality. The results show that SALTMED simulates lower relative yields than the other models for all combinations of irrigation amounts and water quality. For the other models, including SWAP, relative yield values were similar (within about 7% or less) for all irrigation amounts with electrical conductivity below 3 dS/m.

The SWAP user manual and corresponding website describe the theoretical background in detail as well as model input and applications (Kroes et al., 2009). SWAP is developed and maintained by Wageningen University and Research centre.

## 2.2 WOFOST

The underlying principles of WOFOST have been discussed by van Keulen and Wolf (1986). The initial version was developed by the Centre for World Food Studies in Wageningen (van Diepen et al., 1989). The basic processes simulated by WOFOST are phenological development, biomass growth, its partitioning over plant organs, root growth and the soil water balance. The most important external drivers are daily weather data. Other external drivers are initial soil and crop conditions. The most important internal driver is the leaf area index (LAI) which is the result of the leaf area dynamics controlled by photosynthesis, allocation of biomass to leaves, leaf age and development stage. In turn, LAI controls the daily rates of photosynthesis and evapotranspiration.

Currently, WOFOST as described by Boogaard et al. (1998) and Kroes et al. (2009) is able to simulate potential production as governed by atmospheric conditions and plant characteristics, and limited production due to water, oxygen and/or salinity stress. Figure 4 shows the processes and relations incorporated in WOFOST. The radiation energy absorbed by the canopy is a function of incoming radiation and crop leaf area. Using the absorbed radiation and taking into account photosynthetic leaf characteristics, the potential photosynthesis is calculated. The latter is reduced due to water, oxygen and/or salinity stress, as quantified by the relative transpiration ($T_a/T_p$), and yields the actual photosynthesis.

Part of the carbohydrates ($CH_2O$) produced are used to provide energy for the maintenance of the living biomass (maintenance respiration). The remaining carbohydrates are converted into structural matter. In this conversion, some of the weight is lost as growth respiration. The dry matter produced is partitioned among roots, leaves, stems and storage organs, using partitioning factors that are a function of the crop development stage. The amount partitioned to the leaves determines leaf area development and hence the capacity of light interception. This interaction of light interception and leaf area growth is a very important positive feedback in WOFOST. The dry weights of the various plant organs are obtained by integrating their growth rates over time. During the development of the crop, part of the living biomass dies due to senescence.

Changes in $CO_2$ directly affect photosynthesis. Rising levels of $CO_2$ will result in higher $CO_2$ uptake but also in closing of the stomata which then reduces $CO_2$ uptake. The net effect however is increase in crop growth. Kroes and Supit (2011) simulated grassland growth with SWAP-WOFOST for several climate scenarios and found that the increase of $CO_2$ concentration is more important than the predicted increase in temperature for both potential and actual yield.

In relation to climate change heat stress may become just as important as drought stress for limiting crop production. It is well known that short episodes of high temperatures during the flowering period can drastically reduce the productivity of many field crops. However, the version of the WOFOST crop simulation model currently integrated in SWAP does not consider the direct impact of heat stress on grassland or maize productivity. For the moderate ocean climate of the Netherlands it is not expected that the direct impact of heat stress will lead to considerable yield losses in the coming decades (Teixeira et al., 2013). Furthermore, the amount of experimental data to parametrize such relationships is limited which has hampered adding such algorithms with sufficient confidence in WOFOST. It is only recently within the framework of the AgMIP project that the systematic testing of models against dedicated heat stress experiments is taking place (Liu et al., 2016).

**2.3 Linked models and model testing**

We have linked these two simulation models SWAP and WOFOST on a daily basis to ensure realistic interaction between water in the root zone and crop growth. This interaction allows for dynamic root growth as a function of weather and soil conditions, dynamic crop growth as a function of weather, crop characteristics and water availability and also more realistic

calculation of transpiration as a function of dynamic crop cover and leaf area simulations. Because both models are process based, the linkage of these models enables us to assess the effects of future climate on the interaction between hydrology and crop growth (Bartholomeus et al., 2012; Guisan & Zimmermann, 2000). In the current SWAP-WOFOST version two production situations are simulated: the potential and water-limited situation. The potential crop production situation is

defined by temperature, day length, solar radiation and crop characteristics. Optimum nutrient and moisture levels are assumed. The water-limited situation is defined by the above mentioned factors in combination with water or oxygen shortages.

For testing the performance of the combined SWAP-WOFOST model we needed data from experiments where both

hydrological parameters and crop growth were measured and where no other limitations for crop growth occurred than water-related limitations, i.e. drought or too wet conditions. Kroes et al (2015) described how for this purpose three datasets remained for grassland and two datasets for silage maize. For grassland data from a sandy soil (*Ruurlo*) and a peat soil (*Zegveld*) were available and for silage maize we had access to data from two different experiments on sandy soils, one of which in a relatively dry situation (*Cranendonck*) and the other with higher groundwater levels and a more loamy texture

(*Dijkgraaf*). The Ruurlo data were available for 1980-1984 (two different fields) and the Zegveld data for 2003-2005. Then the Cranendonck data for silage maize were measured in the years 1974-1982 and the Dijkgraaf data were more recent (2007-2008).

The grassland composition of the experimental sites that were used for calibrating SWAP-WOFOST mainly consists of

varieties of English ryegrass (*Lolium perenne*) which is the dominant type of grassland in the Netherlands and Western Europe. See also Schapendonk et al (1998) for a description of similar sites that were used for the related LINGRA model.
The crop files for silage maize are based on the standard WOFOST crop files for maize that are calibrated for Netherlands and Germany. These crop files are based on field trials executed in Belgium, United Kingdom and the Netherlands and on research executed in the framework of the MARS project (https://ec.europa.eu/jrc/en/mars). Typical heat sums for silage

maize are 750 degree days for emergence to flowering and 850 degree days for the period from flowering to ripeness with a baseline temperature of 8 degrees Celsius.

### 2.4 Meta-model

The definition of a meta-model in WaterVision Agriculture is that a meta-model is a model derived from another model. In the case of the linked SWAP-WOFOST model this means that the meta-model of SWAP-WOFOST must be able to simulate

crop growth as if it was directly calculated using SWAP-WOFOST. A meta-model thus models the model results from another model (the original model).

A meta-model is usually a lot less complex than the original model. This can be explained because a meta-model only describes a small part of the original model. In the case of WaterVision Agriculture we are looking for a meta-model that can reproduce the annual average crop yield reduction as a function of drought, too wet or too saline conditions. All other model results simulated by SWAP-WOFOST, like water content in the root zone or daily biomass production will not be addressed

by the meta-model.

The advantage of having a meta-model is that it requires much less input data than the original model. For SWAP-WOFOST simulations for instance we need a soil profile description with hydraulic characteristics and a large number of crop characteristics. For using the meta-model we only need to know soil type and crop type. This will make the meta-model a lot

easier to use and it speeds up the calculations. So, based on the complex process-based model SWAP-WOFOST easily applicable statistical relationships have been derived between groundwater characteristics and crop yield. These relationships, the meta-model, mimic the relevant processes involved and generate roughly the same model results as the SWAP-WOFOST model would do.

The meta-models we use for WaterVision Agriculture are so called random forests (Breiman, 2001). Random forests consist of many (usually several hundreds of) classifications or regression trees (CART-models). In our case, we have grown forests with regression trees. Each regression tree predicts crop growth given a set of explanatory variables like crop type, soil type, meteorological district, climate scenario, and several groundwater characteristics (e.g. mean groundwater level, mean highest groundwater level, mean lowest groundwater level, average spring groundwater level). Starting at the trunk of a regression

tree, the data are recursively split into smaller parts based on simple rules like "IF soil type is sand THEN follow the left branch up the tree ELSE follow the right branch up the tree". Each branch of the tree is split in turn until a terminal leaf is reached. This leaf contains a prediction (in our case crop growth). Instead of a single tree, random forests employ an entire ensemble of regression trees (forest of trees) to improve prediction accuracy by averaging the predictions of all individual regression trees.

**2.5 Input data for deriving the meta-model**

For deriving the meta-model, the SWAP-WOFOST combination was run approximately 360,000 times. This number is a result of simulation runs for two crops (grassland and silage maize), 72 soil units of the Dutch soil physical database, five weather stations, current weather and four climate scenarios. As lower boundary condition for the SWAP model we used the $q_b(h)$-relation (Kroes et al., 2009). This relation assumes that the vertical flux ($q_b$) is related to groundwater level ($h$)

according to:

$q_b = A \, exp \, (B \, h) + C$

where *A, B* and *C* are coefficients. For deriving the meta-model we assume that *A* can vary between -10 to 0, *B* between -0.10 to -0.01 and finally *C* is related to the boundary of drainage (*zd*):

*C = -A exp (B abs(zd))*

Where *zd* can vary between 250 to 25 cm below soil surface.

Meteorological data were available from the Dutch meteorological institute KNMI. This involves daily global radiation, minimum and maximum temperature, air humidity, wind speed, rainfall amounts and duration for five weather stations in the
Netherlands for 30-year periods. KNMI provides current weather data for the period 1981-2010 as well as projected data for 30-year periods around 2050 for different climate scenarios (KNMI, 2014). Crop yield reduction has been simulated for two crops: grassland and silage maize.

Soil profile information was obtained from the BOFEK 2012 data-base (Wösten et al., 2013). It contains soil physical data
for 72 representative soil profiles covering the whole of the Netherlands. For each combination of crop (2x), soil profile (72x), weather station (5x), and climate scenario (5x), 100 sets of boundary conditions have been drawn by means of Latin hypercube sampling (Iman & Conover, 1982). This sampling method enforces an efficient coverage of the parameter space.

### 2.5.1 Soil physical data BOFEK

Soil water transport in the unsaturated zone is largely affected by the soil hydraulic characteristics (water retention curve pF
and hydraulic conductivity $k(h)$). Such data are available for the whole of the Netherlands from the national soil physical database BOFEK (Wösten et al., 2013). This database has 72 soil profiles with a vertical soil layer schematization that is characterised by 36 different soil physical relations for top soil and subsoil layers.

Using these soil characteristics tests were carried out with the SWAP model for these 72 soil physical units and different
types of land use using different input options (tabular input of pF- and $k(h)$-functions or using Mualem-Van Genuchten-parameters (Mualem, 1986; Van Genuchten, 1980)). This resulted in small adjustments in the numerical solutions of the SWAP model to increase calculation speed and in small changes in the database to eliminate minor errors. An improved version of the database was made available on the website (http://www.wageningenur.nl/nl/show/Bodemfysische-Eenhedenkaart-BOFEK2012.htm ).

## 3 Results

### 3.1 Test results linked SWAP-WOFOST

The linked SWAP-WOFOST model was evaluated using five experimental datasets with observations for grassland and silage maize (Table 1). These sets were selected because the experiments had a focus on stress due to drought or wet conditions; other stresses, like nutrient shortage or pests and diseases hardly occurred at these experiments which allowed an evaluation of the SWAP-WOFOST model for water stress situations. The experimental data of these five sets come from quite a range of studies, performed in different years and different parts of the country. For all these sets simulation studies have been performed before with the SWAP model and we assume calibration has taken place. The SWAP-related input data and parameters that were used before have also been used now. For crop growth the standard parameter sets for WOFOST were used. A detailed analysis of the hydrological conditions is given by Kroes et al. (2015); in this paragraph a summary of the crop yield results is presented because we regard crop production as the most relevant indicator for the performance of the linked SWAP-WOFOST model.

For the grassland sites the yields of the different grassland cuts were compared with the observed values (Fig. 5). The mean error between annual observed and simulated yield for the three grassland sites is respectively 2.8, 2.7 and 0.2 ton/ha dry matter. We think that the agreement between observations and simulations is rather satisfactory. Annual simulated grassland yields were general higher than the observed annual yields. That is especially true for the year 1982 at the two Ruurlo experiments, where actual and observed results of the years 1980, 1981 and 1984 compare well and the results of the year 1982 shows the largest difference between simulated and observed. A more detailed analysis explained this overestimation as an effect of simplifications in model approaches and also the impact of the cold spring seems to be simulated not accurately enough. At the Ruurlo site only drought stress occurs; there is hardly ever excess of water at this sandy soil. At the organic soils of Zegveld both too dry and too wet conditions can occur. Especially 2003 was a dry year.

The two sites with experimental data for silage maize were used for evaluating SWAP-WOFOST by comparing the simulated and observed total above ground biomass at harvest (Fig. 6). The mean error between annual observed and simulated yield shows an underestimation of respectively 2.8 and 3.6 ton/ha dry matter. We regarded the simulated yields in the different years at the Cranendonck site and the single experiment at Dijkgraaf as satisfactory given the uncertainties and simplifications of the model approaches. At Cranendonck drought stress is the main factor influencing productivity. At Dijkgraaf both too dry and too wet conditions may occur, but for this site we only have data for one year. Both stresses occurred in that year with a dry spring and relatively wet summer months. The drought stress in the extremely dry year of 1976 at Cranendonck was approached relatively well, which we regard as an indicator of the ability of the combination of SWAP and WOFOST to enable the simulation of extreme events.

## 3.2 Examples for application of SWAP-WOFOST

SWAP-WOFOST generates insight in the variation in relative transpiration and in crop yield on different time scales and for different climate scenarios, each relevant for different user questions. Figure 7 (A and C) shows annual variation in transpiration reduction due to either too dry or too wet conditions for both current (A) and future climatic conditions (C). Based on these simulations climate average (30-year) values can be derived, which are relevant for quantifying the direction of changes as related to long-term changes in climate conditions or in water management. Figure 7 A illustrates that the climate-average drought stress can be relatively minor, but that peaks in stress can occur in specific years, like the year 2003. SWAP-WOFOST allows to analyse such years in more detail, with a focus on extreme events (Fig. 7 B). Due to climate change drought stress may increase significantly (Fig. 7 C and D), while oxygen stress shows only a minor increase. Figure 7 D also shows that even in a very dry year, an abundant rainfall event may occur in another period of the year which also results in reduced crop growth. The different levels of application serve different needs, from policy making (long-term averages) to operational water management (daily values).

## 3.3 Examples of the meta-model WaterVision Agriculture

Figure 8 shows examples of the meta-model for 9 different soil types of the BOFEK database, for grassland and the current climate in De Bilt, with the mean highest groundwater level (MHG) on the y-axis and the mean lowest groundwater level (MLG) on the x-axis. Red dots represent crop yield reduction due to drought and blue dots are the result of situations with too wet conditions for crop growth. The size of the dots indicate the average annual amount of crop yield reduction.

An important goal of the WaterVision Agriculture project is enabling the application of the meta-model to any area in the Netherlands. As an example we applied the meta-model to an area in the south of the Netherlands with mainly sandy soils. The location of this area is shown in Fig. 9. This example area is an area where drinking water is pumped up and the influenced area is considered to be almost circular. Figure 10 shows the effect on crop yield resulting from the lowering of the groundwater levels as annual average percentage (as the sum of dry and wet conditions). The differences on the map are mainly caused by differences in soil type. The changes in crop growth are largest near the drinking water well.

## 4 Discussion

The project Watervision Agriculture aims at a climate-proof instrument that can determine crop yield effects as a result of drought, too wet or too saline conditions, based on process based models. Furthermore the instrument has to be applicable to various crop and water management situations as required by the end users. It can be used at field level and for evaluating crop yield as a function of current weather and also extreme weather events as described in section 3.2 and the meta-model allows a quick application on regional level showing long term effects of hydrological measures in section 3.3. With these

different components of WaterVision Agriculture we have made a toolbox for different applications as requested by the different end users.

In the next phase of the project similar work for other crops than grassland and silage maize is scheduled as well as a method to add farm economic effects to the evaluation tools. Furthermore in the coming year the indirect effects of drought and oxygen stress will be addressed. This includes for instance the effect of too wet conditions for harvest, resulting in yield losses or damage to the soil structure when harvest takes place anyway. Indirect effects are also related to crop quality or postponing grazing or cutting of grassland. Furthermore it is expected that new information will become available on salt tolerance levels of different crops. This information should then also be included in WaterVision Agriculture. For the future some issues remain that may get attention in a following project. Many users would like to include nutrient effects on crops, which also allows an evaluation of fertilization and groundwater quality.

## Author contribution

Mirjam Hack-ten Broeke is an expert on agrohydrology and soil science at Alterra and has specialized in project management. She is the project manager of the Watervision Agtriculture project.

Joop Kroes is an expert on soil hydrology and on the SWAP model at Alterra. He is responsible for the model combination of SWAP and WOFOST.

Ruud Bartholomeus is an expert in agrohydrology and climate change at KWR and is responsible for the valuable addition of the process-based simulation of oxygen stress.

Jos van Dam is an expert on agrohydrology and soil physics at Wageningen University and shares the responsibility for the SWAP model.

Allard de Wit and Iwan Supit are experts on the WOFOST model at Alterra and Iwan Supit especially in relation to grassland.

Dennis Walvoort is an expert on physical geography and spatial statistics at Alterra and has contributed the meta-model to the project.

Jan van Bakel is an expert on agrohydrology at De Bakelse Stroom. He helped starting up the project and has contributed to the Watervision Agriculture tools and the modelling exercises through his long experience in agronomy and hydrology.

Rob Ruijtenberg is the project manager at STOWA and is the coordinator of the end user contributions.

## Acknowledgements

This project is financed by a large group of financers: STOWA (Applied Research of the Water Boards), Ministry of Infrastructure and Environment, ACSG (Advisory Commission for Damage related to Groundwater), provinces Utrecht and

Zuid-Holland, ZON (Zoetwatervoorziening Oost-Nederland), Water companies Vitens and Brabant Water, VEWIN, LTO and the Ministry of Economic Affairs (project KB-14-001-046).

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

Wösten, Henk; de Vries, Folkert; Hoogland, Tom; Massop, Harry; Veldhuizen, Ab; Vroon, Henk; Wesseling, Jan; Heijkers, Joost; Bolman, Almer, 2013.  BOFEK2012, de nieuwe bodemfysische schematisatie van Nederland. Alterra, Wageningen. Rapport 2387 (in Dutch).

**Figure captions**

Figure 1. Transport processes and modelling domain of SWAP.

Figure 2 a. Transpiration reduction factor $\alpha_s$ as function of soil water pressure head and b. Transpiration reduction factor $\alpha_s$ as function of soil water electrical conductivity.

Figure 3. Schematization of the oxygen module used to simulate daily respiration reduction. The model combines interacting physiological processes (i.e. root respiration and microbial respiration) and physical processes (i.e. macro-scale and micro-scale oxygen diffusion). Details of equations involved are given in Bartholomeus et al. (2008).

Figure 4. Flow chart of crop growth processes included in WOFOST

Figure 5. Results of simulated and observed yields of grassland: Ruurlo16 (upper figure), Ruurlo48 (middle figure), and Zegveld03 (lower figure). The green lines correspond with simulated potential yield; blue with the simulated exploitable yield; black with the simulated actual yield; and the red dots indicate the observed yield of a grassland cut. SIMmean, OBSmean and ME are annual mean values for simulated actual yield, observed yield and the difference (maximum error ME).

Figure 6. Results of simulated and observed yields of silage maize: Cranendonck 16 (upper figure) and Dijkgraaf (lower figure). The green lines correspond with the simulated potential yield; blue with the simulated exploitable yield; black with the simulated actual yield; red dots indicate the observed dry matter yield. SIMmean, OBSmean and ME are annual mean values for simulated actual yield, observed yield and the difference (maximum error ME).

Figure 7. SWAP-WOFOST simulations of transpiration reduction due to drought stress (Treddry) and oxygen stress (Tredwet) for a silage maize crop on a fictitious sandy soil. The panels show both the different causes of stress and the different time scales the model can be used for. A: yearly cumulative transpiration reduction due to drought stress and the climate-average (30-year) drought stress (horizontal line). B: potential and actual transpiration for 2003. The red polygon, representing the difference between potential and actual transpiration, demonstrates the period and level of drought stress. C-D: idem A-B, but for oxygen stress instead of drought stress.

Figure 8. Meta-model WaterVision Agriculture: examples for 9 different soil types of the BOFEK database (SMU=Soil Mapping Unit) for grassland and the current climate in De Bilt, with the mean highest groundwater level (MHG) on the y-axis and the mean lowest groundwater level (MLG) on the x-axis. Red dots represent crop yield reduction due to drought and blue dots represent crop yield reduction as a result of too wet conditions (oxygen stress).

Figure 9. Location and topography of the Vierlingsbeek area in the Netherlands. The black triangle is the location of the drinking water well.

Figure 10. Application of the meta-model WaterVision Agriculture to the Vierlingsbeek area in the Netherlands. The effect on crop yield resulting from the lowering of the groundwater levels is shown as annual average increase in yield reduction compared to a situation without changes in groundwater levels.

**Tables**

*Table 1 Testsets for SWAP-WOFOST*

| Nr | Crop | Location | Period | Soil type | Reference |
|---|---|---|---|---|---|
| 1 | Grassland | Ruurlo16 | 1980-1984 | Cambic Podzol | Kroes and Supit (2011) |
| 2 | Grassland | Ruurlo48 | 1980-1984 | Cambic Podzol | Kroes and Supit (2011) |
| 3 | Grassland | Zegveld03 | 2003-2005 | Terric Histosol | Hendriks et al. (2011) |
| 4 | Silage maize | Cranendonck16 | 1974-1982 | Cumulic Anthrosol | Schröder (1985) |
| 5 | Silage maize | Dijkgraaf | 2007-2008 | Umbric Gleysol | Elbers et al. (2010) |

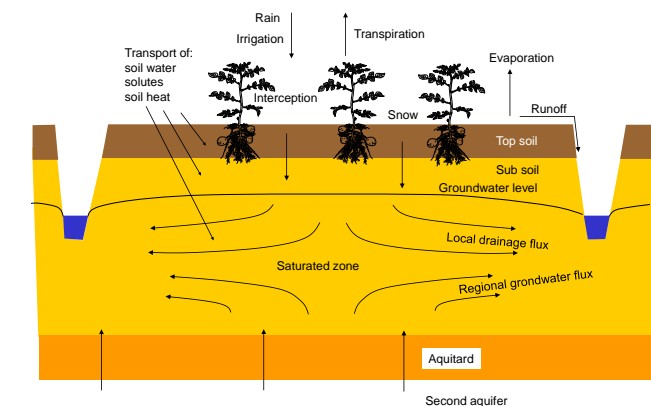

Figure 1. Transport processes and modelling domain of SWAP.

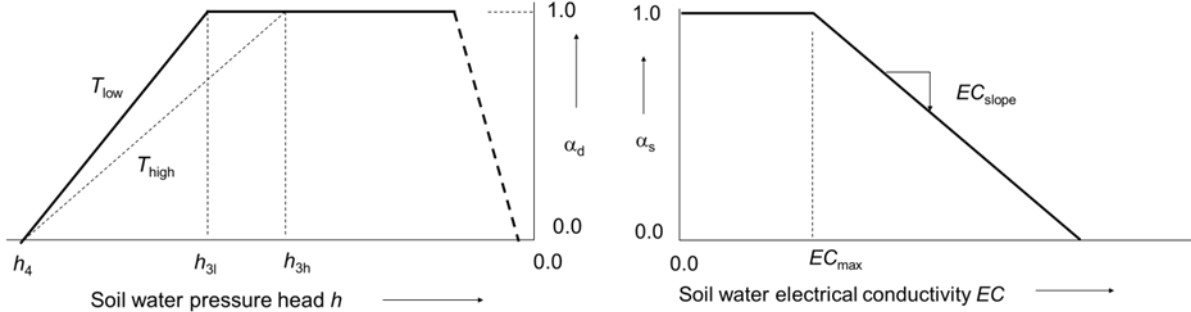

Figure 2a. Transpiration reduction factor $\alpha_s$ as function of soil water pressure head.

Figure 2b. Transpiration reduction factor $\alpha_s$ as function of soil water electrical conductivity.

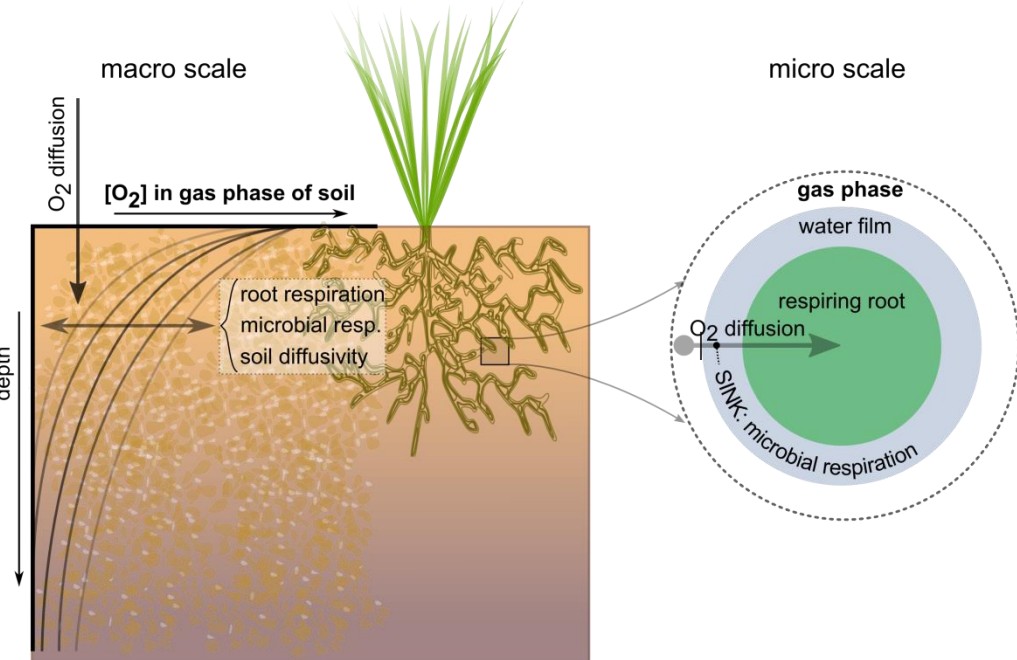

5    Figure 3. Schematization of the oxygen module used to simulate daily respiration reduction. The model combines interacting physiological processes (i.e. root respiration and microbial respiration) and physical processes (i.e. macro-scale and micro-scale oxygen diffusion). Details of equations involved are given in Bartholomeus et al. (2008).

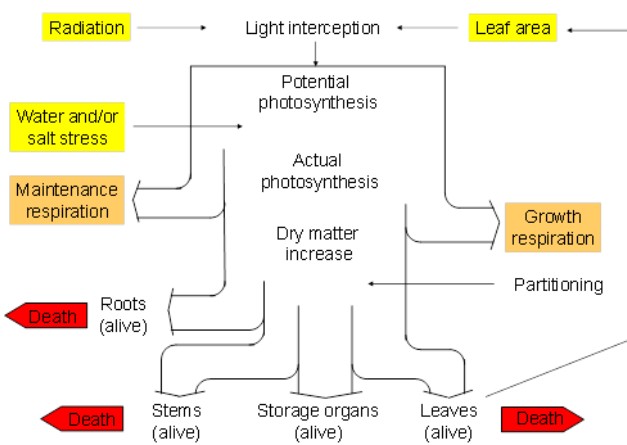

Figure 4. Flow chart of crop growth processes included in WOFOST

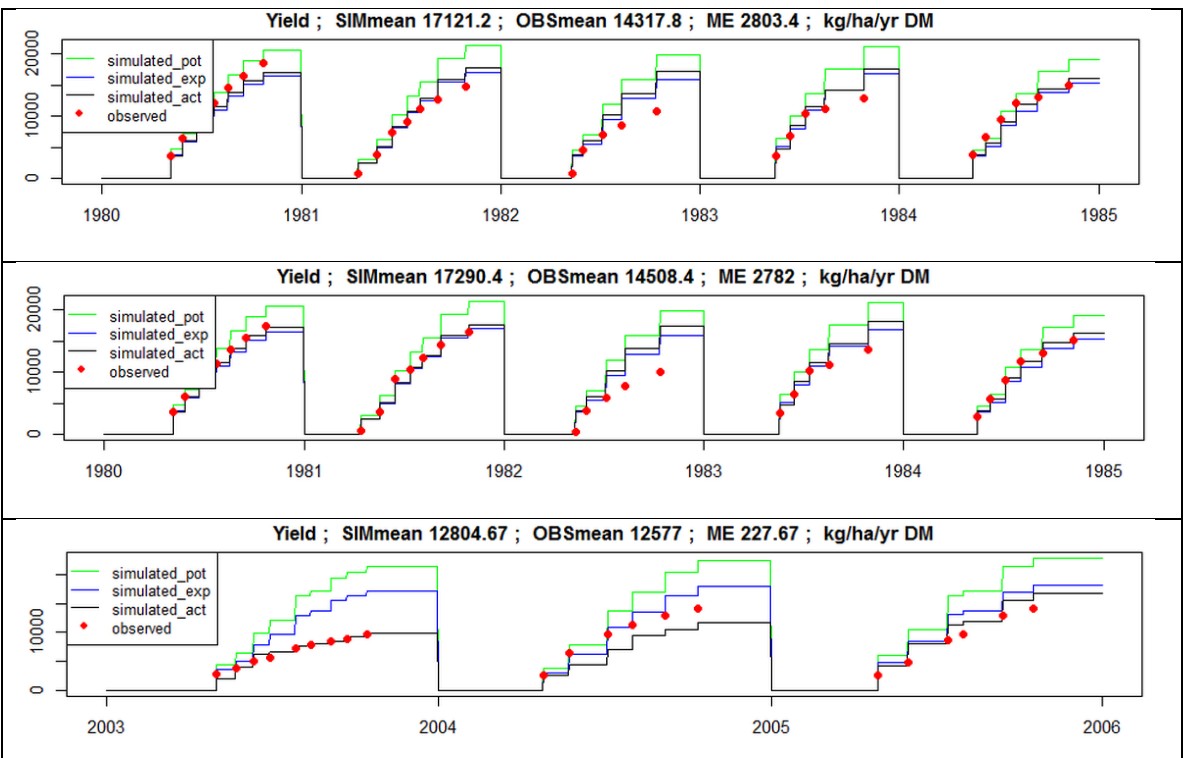

Figure 5. Results of simulated and observed yields of grassland: Ruurlo16 (upper figure), Ruurlo48 (middle figure), and Zegveld03 (lower figure). The green lines correspond with simulated potential yield; blue with the simulated exploitable yield; black with the simulated actual yield; and the red dots indicate the observed yield of a grassland cut. SIMmean, OBSmean and ME are annual mean values for simulated actual yield, observed yield and the difference (maximum error ME).

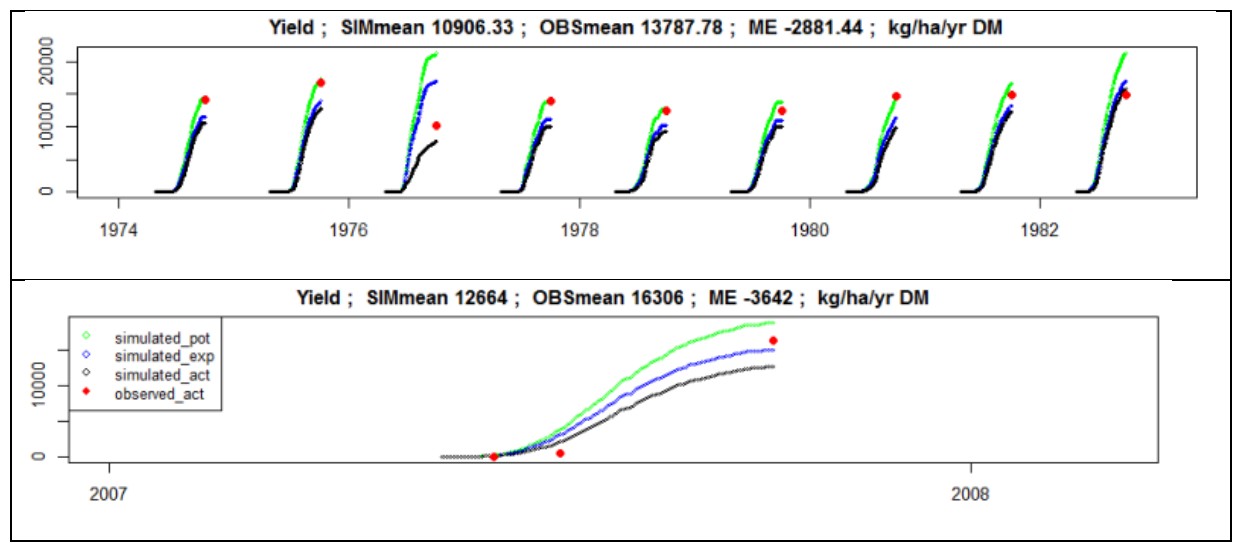

Figure 6. Results of simulated and observed yields of silage maize: Cranendonck 16 (upper figure) and Dijkgraaf (lower figure). The green lines correspond with the simulated potential yield; blue with the simulated exploitable yield; black with the simulated actual yield; red dots indicate the observed dry matter yield. SIMmean, OBSmean and ME are annual mean values for simulated actual yield, observed yield and the difference (maximum error ME).

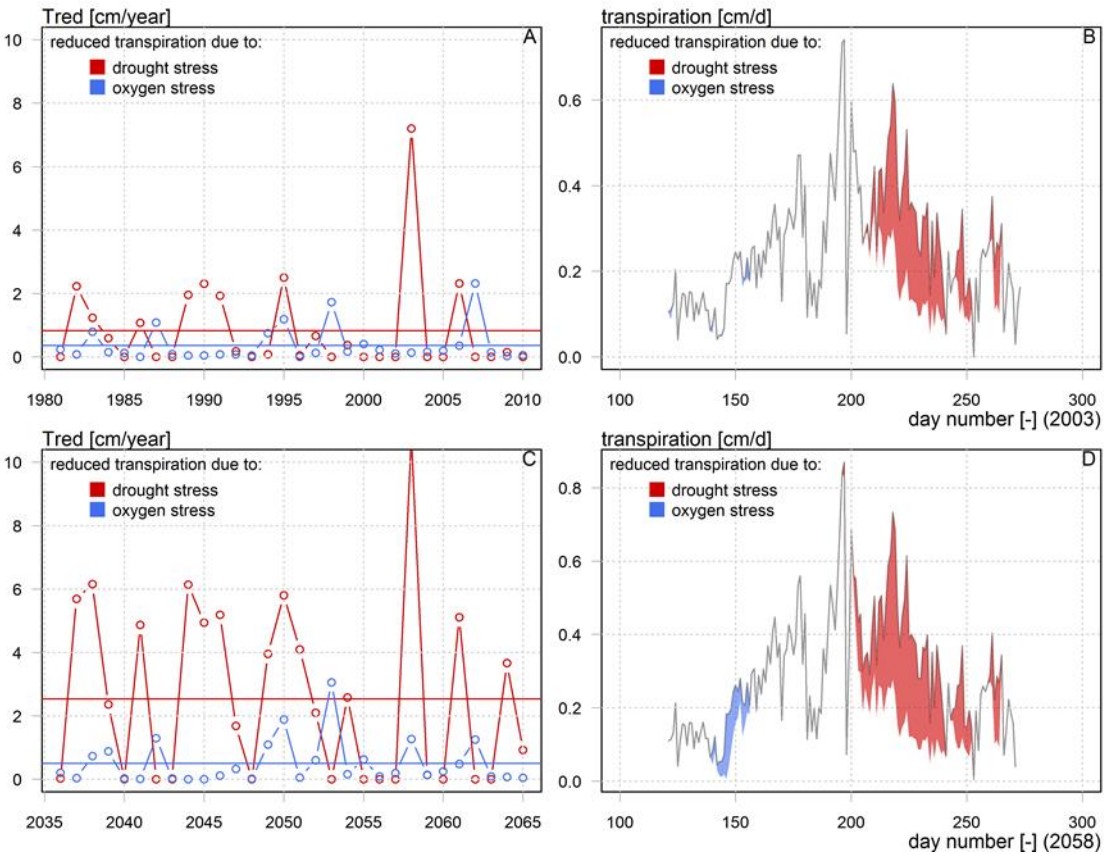

Figure 7. SWAP-WOFOST simulations of transpiration reduction (Tred) due to drought stress and oxygen stress for a silage maize crop on a fictitious sandy soil. The panels show both the different causes of stress and the different time scales the model can be used for. A: yearly cumulative transpiration reduction due to drought stress and oxygen stress and the climate-average (30-year) stresses (horizontal lines). B: potential and actual transpiration for 2003. The red and blue polygons, representing the difference between potential and actual transpiration, demonstrates the period and level of drought stress and oxygen stress. C-D: same as A-B, but for future climate conditions instead of current climate, using climate scenario Wh of the KNMI, representing 2°C global temperature rise and a high value for change in air circulation patterns (KNMI, 2014).

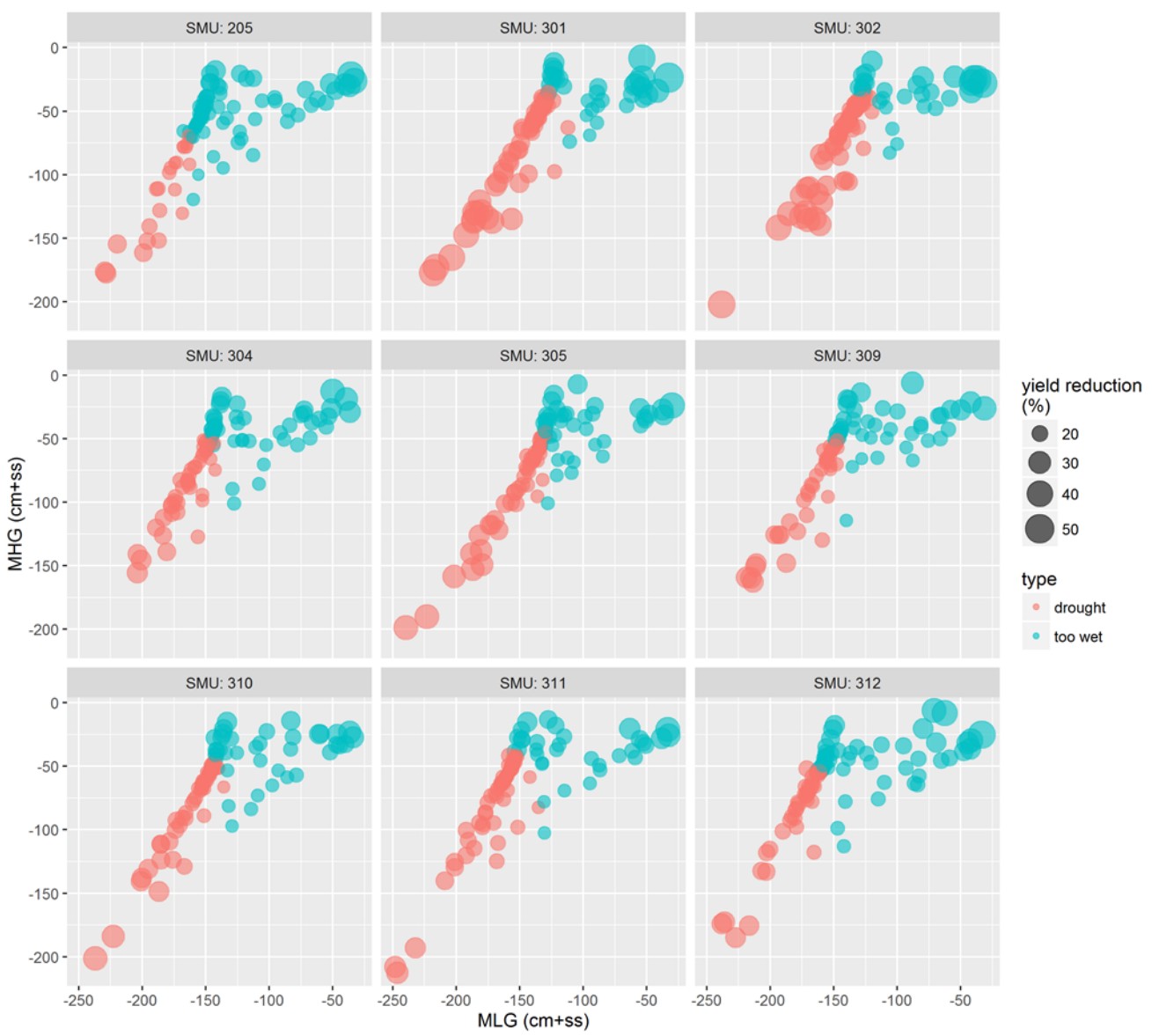

Figure 8. Meta-model WaterVision Agriculture: examples for 9 different soil types of the BOFEK database (SMU=Soil Mapping Unit) for grassland and the current climate in De Bilt, with the mean highest groundwater level (MHG) on the y-axis and the mean lowest groundwater level (MLG) on the x-axis. Red dots represent crop yield reduction due to drought and blue dots represent crop yield reduction as a result of too wet conditions (oxygen stress).

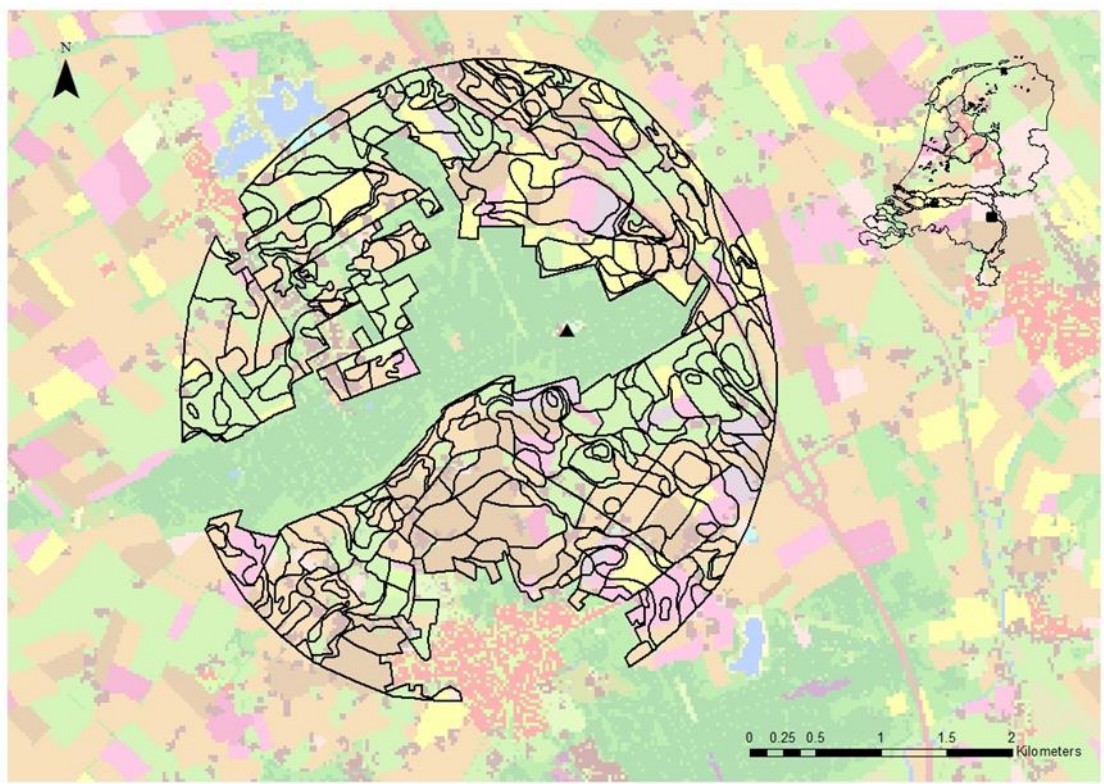

Figure 9. Location and topography of the Vierlingsbeek area in the Netherlands. The black triangle is the location of the drinking water well.

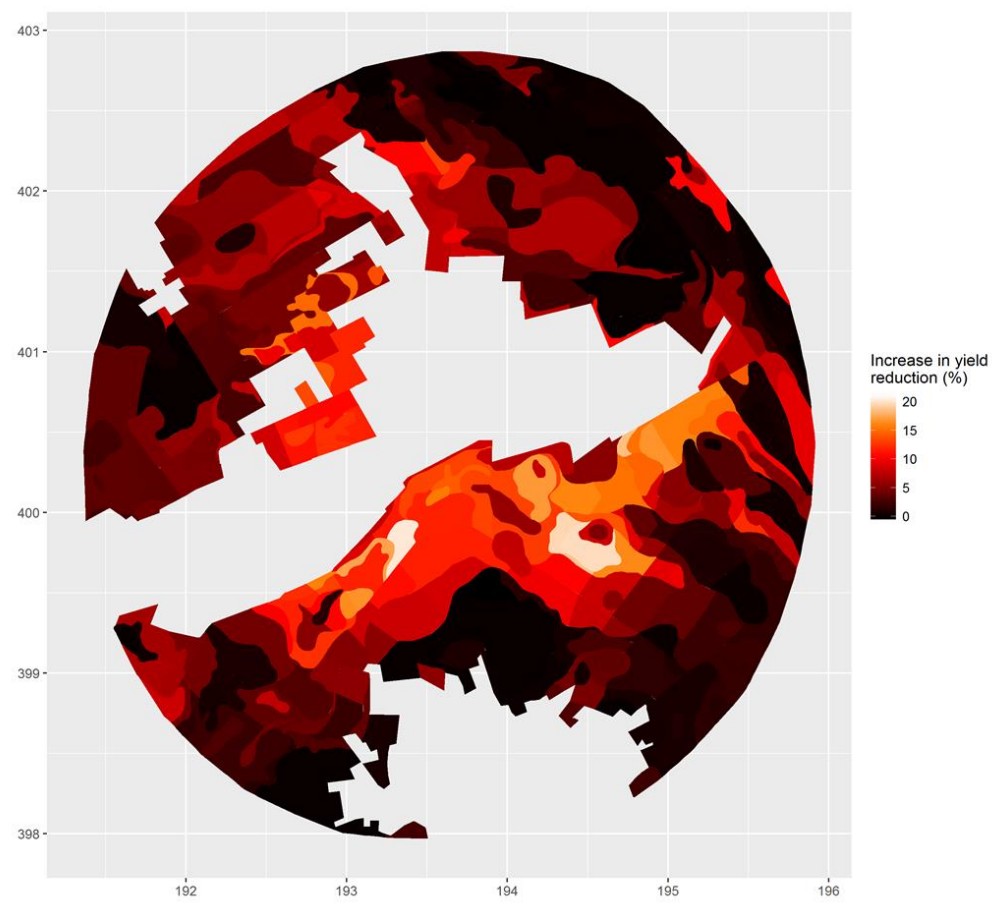

Figure 10. Application of the meta-model WaterVision Agriculture to the Vierlingsbeek area in the Netherlands. The effect on crop yield resulting from the lowering of the groundwater levels is shown as annual average increase in yield reduction compared to a situation without changes in groundwater levels.

