# Peer review of "Quantification of the impact of hydrology on agricultural production as a result of too dry, too wet or too saline conditions"

_SOIL, 2016_

## Referee Comment (RC1) · Anonymous Referee #1 · 10 May 2016

The quantitative assessment of management and hydrological measures on agricultural production is a very interesting topic for end users. Process-oriented modelling is a widely applied measure for such assessments. However, the transfer of model results to end users is a challenge due to the complex interactions in the soil-plant-atmosphere system. The authors describe a new way using meta-modelling as an easy-to-use tool for different end user groups to assess quantitatively the effects under current and future climate conditions. The approach has a sound scientific background and the objectives are clearly defined. The authors provide some examples of application to assess effects on crop production at different scales which documents the range of applicability of the method. For the assessment three main mechanisms of crop yield

reduction were considered: drought, saline conditions and water logging induced oxygen shortage. Although the model approach for saline conditions is described for the process-oriented model there are no results presented for this stress condition. The paper is generally well written and clearly structured. The title and abstract are informative. In the introduction some references about the importance of the three stress conditions in NL, EU and word wide would be desirable (e.g., Jones et al., 2012 EEA State of Soil in Europe). In section 3.1 the SWAP-WOFOST model has been "evaluated" with five experimental data sets. From the paper it is not clear if the model was calibrated for each location separately of if the model was just applied using standardized values. This would be good to know to judge if the mean errors are acceptable. I suggest to add few sentences relating the mean errors to other findings in the literature for calibrated or "blind test" model applications (e.g. published model intercomparisons), respectively. In this section a few comments which stresses became relevant at each site would also be helpful, since the given reference (Kroes et al. 2015) is only available in Dutch. The section 3.3 would require a little more text, since presently the text more or less repeats the figure captions. For Fig. 9 some explanations regarding the area without colors would be useful. The position of the well could be added to the figure as well. Finally, it would be helpful to provide some comments about the reasons of the yield diversity within the circle (e.g. related to soil properties etc.) and how the groundwater distance varies across the circle. The reference of "Feddes et al., 1978" (page 5, line 19) needs to be added to the reference list. Please correct the name of "van Genuchten" on page 9, line 25, and add a corresponding reference here and in the list. On page 11, line 4: the term "development" is misleading, please replace with "growth". In Fig. 6 the legend covers the first part of the upper graph. Since the legend is the same as in the graph below, it could be removed.

[Figure]

---

## Referee Comment (RC2) · Anonymous Referee #2 · 17 May 2016

The paper is an innovative and original contribution to spatial crop model application options, in regard to deal with spatial uncertainties finding optimum compromise between model complexity and model input data. The paper describes the combination of SWAP and WOFOST models and based on that the development of an statistical metamodel demonstrated for spatial application under a set of framework conditions (for the Netherlands) including present and future climate conditions with focus on yield effects of drought, salt and oxygen stress in soil. Suggestions for minor improvements: - Crops: Please describe in more detail the composition of the grassland type on which experimental data the calibration was carried out (i.e. which are the dominating plants?). For silage maize provide cultivar details i.e. the temperature sum

requirements. - describe if and how the SWAP-WOFOST considers the direct CO2 effect on crops; is it considered also in the meta model? Discuss potential impacts on drought stress under the climate scenarios. - Beside drought stress more details should be presented on heat stress effects, i.e. if SWAP-WOFOST considers also direct heat effects (i.e. on fertility for maize or grassland grass types) on the selected crops (i.e. on fertility for maize or grassland grass types). This might be important as drought stress can foster direct heat damages of crops, so heat could be the dominating damaging factor. This aspect could be discussed i.e. how it may change under climate change conditions (in Netherlands) and if the calibration data sets did include such expected more extreme conditions to test the simulated crops response?

---

## Author Comment (AC1) · 30 Jun 2016

**Quantification of the impact of hydrology on agricultural production as a result of too dry, too wet or too saline conditions**

5   M.J.D. Hack-ten Broeke[1], J.G. Kroes[1], R.P. Bartholomeus[2], J.C. van Dam[3], A.J.W. de Wit[1], I. Supit[1]
D.J.J. Walvoort[1], P.J.T. van Bakel[4], R. Ruijtenberg[5]

[1]Alterra, Wageningen University and Research Centre, PO Box 47, 6700 AA Wageningen, the Netherlands
10  [2]KWR Watercycle Research Institute, PO Box 1072, 3430 BB Nieuwegein, the Netherlands
[3]Wageningen University, Soil Physics and Land Management group, PO Box 47, 6700 AA Wageningen, the Netherlands
[4]De Bakelse Stroom, Simon Vestdijkstraat 15, 6708 NW Wageningen, the Netherlands
[5]STOWA, postbus 2180, 3800 CD Amersfoort, the Netherlands

15  *Correspondence to*: M.J.D. Hack-ten Broeke (mirjam.hack@wur.nl)

**Author's response**

We would like to thank the reviewers for their positive remarks and their constructive comments. We suggest the additions and changes to the paper as mentioned below (added at **AC:** in the cited responses below). We hope that these contribute to

5 improve the quality of the paper. All changes are highlighted in the text.

In response to Anonymous Referee #1 ()

The quantitative assessment of management and hydrological measures on agricultural production is a very interesting topic for end users. Process-oriented modelling is a widely applied measure for such assessments. However, the transfer of model

10 results to end users is a challenge due to the complex interactions in the soil-plant- atmosphere system. The authors describe a new way using meta-modelling as an easy-to-use tool for different end user groups to assess quantitatively the effects under current and future climate conditions. The approach has a sound scientific background and the objectives are clearly defined. The authors provide some examples of application to assess effects on crop production at different scales which documents the range of applicability of the method. For the assessment three main mechanisms of crop yield reduction were

15 considered: drought, saline conditions and water logging induced oxygen shortage. Although the model approach for saline conditions is described for the process-oriented model there are no results presented for this stress condition.

The paper is generally well written and clearly structured. The title and abstract are informative. In the introduction some references about the importance of the three stress conditions in NL, EU and worldwide would be desirable (e.g., Jones et al., 2012 EEA State of Soil in Europe).

      **AC:** in the introduction we added some references on the importance of water stresses (page 7, line 30 – page 8, line 23); the references were also added in the reference list.

      Some extra information on the performance of the SWAP model for saline conditions is added in section 2.1 (page 11, lines 12-18)

In section 3.1 the SWAP-WOFOST model has been "evaluated" with five experimental data sets. From the paper it is not clear if the model was calibrated for each location separately of if the model was just applied using standardized values. This would be good to know to judge if the mean errors are acceptable.

5        **AC:** a short explanation on calibration and standard parameter sets is added in section 3.1 (page 16, lines 6-10)

I suggest to add few sentences relating the mean errors to other findings in the literature for calibrated or "blind test" model applications (e.g. published model intercomparisons), respectively.

10        **AC:** in the ongoing AgMIP-project model intercomparison is an important component. The WOFOST modellers are involved in this project. We are not aware of model intercomparison studies involving linked models for (agro)hydrology and crop growth with the same level of detail as the linked SWAP-WOFOST tool. Therefore we suggest not to add text on this topic, for model intercomparison for other types of simulation models may have no meaning in our case.

In this section a few comments which stresses became relevant at each site would also be helpful, since the given reference (Kroes et al. 2015) is only available in Dutch.

       **AC:** for the grassland sites some information was added in section 3.1, page 16, lines 21-22, and for the sites with
20        silage maize in lines 28-30.

The section 3.3 would require a little more text, since presently the text more or less repeats the figure captions. For Fig. 9 some explanations regarding the area without colors would be useful. The position of the well could be added to the figure as well. Finally, it would be helpful to provide some comments about the reasons of the yield diversity within the circle (e.g.
25 related to soil properties etc.) and how the groundwater distance varies across the circle.

**AC:** Would it be helpful to add this figure?

[Figure]

**AC:** A few extra lines were added describing Figure 9 in section 3.3 (page 17, lines 22-23)

The reference of "Feddes et al., 1978" (page 5, line 19) needs to be added to the reference list. Please correct the name of "van Genuchten" on page 9, line 25, and add a corresponding reference here and in the list. On page 11, line 4: the term "development" is misleading, please replace with "growth". In Fig. 6 the legend covers the first part of the upper graph. Since the legend is the same as in the graph below, it could be removed.

**AC**: references were added in the reference list; development was changed into growth (page 17, line 11). If required we could change Figure 6 as requested.

The paper is an innovative and original contribution to spatial crop model application options, in regard to deal with spatial

5  uncertainties finding optimum compromise be- tween model complexity and model input data. The paper describes the

combination of SWAP and WOFOST models and based on that the development of an statistical metamodel demonstrated

for spatial application under a set of framework conditions (for the Netherlands) including present and future climate

conditions with focus on yield effects of drought, salt and oxygen stress in soil. Suggestions for minor improvements:

- Crops: Please describe in more detail the composition of the grassland type on which experimental data the calibration was

10  carried out (i.e. which are the dominating plants?). For silage maize provide cultivar details i.e. the temperature sum

requirements.

**AC:** we added information on grassland type and silage maize in section 2.3 (page 13, lines 19-26)

15  - describe if and how the SWAP-WOFOST considers the direct $CO_2$ effect on crops; is it considered also in the meta model?

Discuss potential impacts on drought stress under the climate scenarios.

**AC:** how $CO_2$ affects crop growth is added in section 2.2 (page 12, lines 16-19)

20  - Beside drought stress more details should be presented on heat stress effects, i.e. if SWAP-WOFOST considers also direct

heat effects (i.e. on fertility for maize or grassland grass types) on the selected crops (i.e. on fertility for maize or grassland

grass types). This might be important as drought stress can foster direct heat damages of crops, so heat could be the

dominating damaging factor. This aspect could be discussed i.e. how it may change under climate change conditions (in

Netherlands) and if the calibration data sets did include such expected more extreme conditions to test the simulated crops

25  response?

**AC:** a text on the importance of heat stress is added in section 2.2 (page 12, lines 21-28), with references. The model WOFOST is not yet capable of simulating heat stress effects.

[revised manuscript text omitted]